# A Dynamic Programming Approach to Ecosystem Management

Alessandra Rosso and Ezio Venturino *,†

Dipartimento di Matematica "Giuseppe Peano", Università di Torino, Via Carlo Alberto 10, 10123 Torino, Italy
* Correspondence: ezio.venturino@unito.it; Tel.: +39-011-670-2833
† Member of the INdAM research group GNCS.

**Abstract:** We propose a way of dealing with invasive species or pest control in agriculture. Ecosystems can be modeled via dynamical systems. For their study, it is necessary to establish their possible equilibria. Even a moderately complex system exhibits, in general, multiple steady states. Usually, they are related to each other through transcritical bifurcations, i.e., the system settles to a different equilibrium when some bifurcation parameter crosses a critical threshold. From a situation in which the pest is endemic, it is desirable to move to a pest-free point. The map of the system's equilibria and their connections via transcritical bifurcations may indicate a path to attain the desired state. However, to force the parameters to cross the critical threshold, some human action is required, and this effort has a cost. The tools of dynamic programming allow the detection of the cheapest path to reach the desired goal. In this paper, an algorithm for the solution to this problem is illustrated.

**Keywords:** optimization in graphs; optimal paths; directed graphs; population models; ecoepidemic models; dynamical systems equilibria

**MSC:** 92D45; 94B60; 90B10; 90C39; 92D25; 92D30; 92D40

## 1. Introduction

In mathematical biology [1], one important topic of investigation is represented by population theory, which has a wide range of applications from the ecosystem level down to cells within the organisms. Generally, these entities are modeled via dynamical systems, formulated either by systems of ordinary differential equations, and in the case where space represents an issue, also by partial differential equations. Their analysis relies on the assessment of their equilibria, when possible, and on numerical simulations if the model is too complex. However, an extension of the algorithm to systems exhibiting sustained oscillations could be considered in future works.

From the theoretical analysis, it is usually found that complex population models may exhibit various behaviors, even producing complicated dynamics or even chaotic behavior. This is excluded from our considerations, as it will be apparent from the examples studied below. However, this irregular system behavior does not always appear, and in many other instances, ecosystems attain several equilibrium states that are sometimes linked to each other via transcritical bifurcations. The ultimate system behavior can thus be summarized by a graph in which nodes stand for equilibria and the arcs for the transcritical bifurcations that link them. The transitions from one equilibrium to a neighboring one are obtained by suitable changes in the model parameters, through the crossing of critical thresholds. Thus, starting from the configuration at which the system is currently found, properly acting on these population-related parameters, if at all possible, allows attaining a desirable final state.

For instance, in a situation in which a crop is infested by some kind of damaging agent, be it insects, fungi or plant diseases of any sort, the management would like to attain a pest-free situation in the agroecosystem.

However, this may not always be attained directly, but through a series of steps that involve some of the other model equilibria. In addition, in practical situations, a cost is associated to each transition from one equilibrium to another, as acting on the model parameters involves some kind of external effort.

Clearly, the implementation of these operations should optimize the procedure by minimizing the costs, to make the management economically viable. The important problem of optimization has received great attention in the literature for its economic implications [2,3].

In this paper, we consider the problem of optimizing the total cost of the operations and approach it here as a shortest-path problem, through a dynamic programming procedure. This approach represents the novelty of the paper.

Dynamic programming is an optimization technique proposed originally by Bellman [4–9]. It has also been widely used in many other settings, see for instance [10–16]. Some other applications concern medical problems [17], finance [18] and psychology [19]. In the wider context of operations research, this approach turns out to be very useful even in cases of crisis [20]. The main idea of dynamic programming consists of subdividing the main problem into smaller subproblems. The solution at each stage is incorporated into the next stage. This procedure is based on the optimality principle, stating that the best policy, i.e., the minimum cost path from the initial node to the final node, must also incorporate the best policy starting from an intermediate node to the final one.

As such, this technique applies to the general problem of network flows [21–27]. It represents also a reliable and fast way of partitioning graphs [28], and it is employed in computer science legalization procedures [29].

In this paper, we illustrate this application through some examples related to biological systems already investigated in the literature [30–33].

The main contribution of this work is represented by the fact that, to our knowledge, this is the first application of such techniques in the context of ecology. Instead, it is to be noted that control problems for the management of ecosystems are currently being investigated, see, for instance, the classical book [34], but this is a completely different issue from our approach and moreover, does not address the same problem.

We illustrate here a program for the detection of the optimal path of transitions, starting from a specified node in the graph of the equilibrium points of the dynamical system to attain the desired outcome. The Matlab code takes as input the existing links between nodes and constructs the graph incidence matrix. It also identifies the sequence of the stages originating from the initial configuration. It is adapted to account also for the case in which a path reaches the final node, representing the desired configuration, before the last stage. In such cases the program is devised to automatically generate a sequence of extra dummy nodes linking up at zero cost the current state to the final node.

## 2. Materials and Methods

### 2.1. Networks

Just for the benefit of the reader, we now proceed to refresh some basic concepts, bearing in mind that the latter are well-known definitions constituting widespread knowledge in the field. Thus, the next three subsections contain standard material that widely overlaps the current literature and can be easily found even in specific textbooks, see, e.g., [5–7]. We briefly review the basic terminology in graph theory, before proceeding to the illustration of the algorithm.

A *network* (or *graph*) consists of a set of *vertices* (or *nodes*) $V$ and a set of *arcs* (*links*, *edges*) $E \subseteq V \times V$, i.e., pairs of vertices. An arc connects two nodes of $V$ and is labeled by the nodes it connects.

*Directed arcs* (*indirected arcs*) are given by ordered, respectively, unordered, pairs of vertices. In particular, directed arcs may represent flows through the network, allowed only in one direction. A network is correspondingly named *directed network* if it contains only directed arcs; otherwise, it is an *indirected network*. Any network can be written as a directed network, replacing undirected arcs with pairs of oppositely directed arcs.

A *path* between two nodes is a sequence of distinct arcs connecting the two nodes. When some of the arcs in the network are directed, we can have *directed paths* or *indirected paths*. *Adjacent* vertices are connected by an edge.

A graph can be represented by an *adjacency matrix*. It is a square matrix $A = (a_{i,j})$ whose elements indicate adjacent pairs of vertices in the graph. Here, $a_{i,j} \in \{0,1\}$ and $a_{i,j} = 1$, if and only if $i$ and $j$ are adjacent nodes. An indirected network is described by a symmetric matrix. More generally, the nonzero elements of an adjacency matrix can be replaced with numbers representing the weights of the corresponding arcs.

### 2.2. Minimum Cost Flow Problems

Minimum cost problems are a class of network optimization models. They consider flow through a network with limited arc capacities and cost, or distance, assigned to each arc. Several important problems in practical applications are their subcases, e.g., finding the shortest path problem, the maximum flow, or the transportation and assignment problems in operations research.

A *minimum cost flow problem* is characterized by the following features:

1. The network is directed and connected;
2. At least one node is a supply node, and at least another one is a demand node; all other nodes are transshipment nodes;
3. The network has enough arcs with sufficient capacity to enable all the flow generated at the supply nodes to reach all the demand nodes;
4. The cost of the flow through each arc is proportional to the amount of that flow. Each concrete operation implies an effort that results in positive total costs for each arc of the whole network;
5. Directional flow through an arc is allowed according to the arc direction, and the maximum amount of flow cannot exceed the arc capacity;
6. The objective is to minimize the total cost of shipping the available supply through the network to satisfy the given demand (or, alternatively, to maximize the total profit for the same operation);

Specific features for our situation, referring to the above points, are the following ones:

2. In our application, we take the source node as the current state of the system, and consider a single destination;
4. In our case, the total costs for each arc depend on the type of human operations necessary for driving the system from the current state to one of its adjacent nodes.

### 2.3. Applications

The class of minimum cost flow problems includes a variety of applications that can be solved efficiently. Generally, these applications involve determining a plan for delivering goods from their sources to intermediate facilities (warehouses, processing facilities) to customers. Examples of these kinds of problems can be found in goods, waste or cash distribution and management.

For a directed and connected network containing at least one supply node and one demand node, the variables are denoted as follows:

$$ij = \text{arc from node } i \text{ to node } j$$

$$x_{ij} = \text{flow through arc } ij$$

$$c_{ij} = \text{cost per unit flow through arc } ij$$

$$u_{ij} = \text{arc } ij \text{ capacity}$$

$$b_i = \text{flow generated at node } i$$

$$b_i > 0 \text{ if } i \text{ is a supply node}$$
$$b_i < 0 \text{ if } i \text{ is a demand node}$$
$$b_i = 0 \text{ if } i \text{ is a transshipment node}$$

We can identify a range of special cases:

A model network for a *transportation problem* contains a supply node for each source and a demand node representing each destination. The network includes no transshipment nodes. All the arcs are directed from a supply node to a demand node, and a flow $x_{ij}$ stands for the distribution of $x_{ij}$ units from source $i$ to destination $j$. The cost per distributed unit is identified with the cost per unit of flow. Since there is no upper bound on the individual $x_{ij}$, we fix each arc capacity to $u_{ij} = \infty$ for each $ij$. The goal is to minimize the total distribution cost. It is necessary for the total amount of supply to be equal to or exceed the overall demand.

A *Transshipment problem* is a minimum cost flow problem where $u_{ij} = \infty$ for each $ij$, meaning that arcs can carry any amount of flow. This kind of problem generalizes special cases of transportation problems with intermediate nodes.

For the *maximum flow problem*, we consider a directed and connected network in which all flow originates at one node, called the *source*, and terminates at another node, called the *sink*. All the remaining nodes are transshipment nodes. There are no arcs directed to the source, or from the sink. The objective is to maximize the total amount of flow given arc capacities, measuring it either from the source or to the sink. Concretely, the flow can represent goods through the distribution or supply network of a company, or water through a system of aqueducts, and so forth.

A *shortest path problem* is generally formulated on undirected and connected networks. However, more versions exist, each one fitting for different types of applications.

The model network must contain an *origin node* and a *destination node*, and a non-negative distance associated with each link. The objective is to minimize a quantity that can be interpreted as total distance traveled, total time or total cost of a sequence of activities.

The latter is the kind of model we consider for our application, adapted to find minimum-cost directed paths from the origin to the destination of a directed network. The problem will be approached through a dynamic programming procedure.

*2.4. Dynamic Programming Procedure*

Dynamic programming is a method for finding optimization algorithms that consist of breaking a problem into smaller trivial subproblems and determining an optimal sequence of interrelated decisions for the main problem.

There is no standard formulation for the dynamic programming method—the technique must be adapted to each specific case—but we can summarize the general structure of the compatible problems:

- The problem can be divided into stages, requiring a policy decision at each stage. The optimal sequence to be obtained is composed of the decisions taken;
- A number of states are associated with the beginning of each stage: the states represent the possible conditions of the system at the given stage. A stage can be associated with an arbitrarily large number of states;
- At each stage, the policy decision moves the system from the current state to a state associated with the beginning of the successive stage;
- The solution procedure aims at finding one or more *optimal policies* for the main problem; that is, a set of optimal policy decisions at each stage, for each of the possible stages—meaning an indication of the best decisions available under any circumstances;
- The problem must fit the *principle of optimality*: given a state, an optimal policy for the remaining stages is independent of the policy decisions adopted in the previous stages. Therefore, the optimal immediate decision depends only on the current state, regardless of the chosen path;
- The solution procedure begins by finding the optimal policy for the last stage, for each of the possible states, and the subproblems related to each stage are usually trivial or quasi-trivial;

- The optimal policy for a stage n, given the policy for the stage $n + 1$, is identified by a recursive relationship. Its precise form depends on the specific problem, but we can set the following notation to represent the general relationship:

$$N = \text{number of stages}$$

$$n = \text{label for current state}$$

$$S_n = \text{current state for stage } n$$

$$x_n = \text{decision variable for stage } n$$

$$x_n^* = \text{optimal value of } x_n \text{ given } S_n$$

$f_n(S_n, x_n)$ = contribution of stages $n, n+1, \ldots, N$ to the objective function if the system starts at state $S_n$ at stage $n$ (the immediate decision is expressed by $x_n$, and optimal decisions are made afterward).
$f_n^*(S_n) = f_n(S_n, x_n^*)$, recursive relationship for the optimal value function, it is always one of the two forms:

$$f_n^*(S_n) = \min_{S_n+1} f_n(S_N, x_n^*), \quad f_n^*(S_n) = \max_{S_n+1} f_n(S_N, x_n^*),$$

with $f_n(S_n, x_n)$ written in terms of $S_n, x_n, f_{n+1}^*(S_{n+1})$ and eventually a measure of the immediate contribution of $x_n$ to the objective function;
- The solution procedure follows the recursive relationship starting at the end and moving backward stage by stage, finding the optimal policies for the partial problem at each stage. It stops at the initial stage, completing the solution for the entire problem.

Dynamic programming procedures can be adapted to fit a variety of models. *Deterministic dynamic programming* is associated with problems whose state at the next stage is entirely determined by the state and policy decision at the current stage, while in *probabilistic dynamic programming*, a probability distribution describes the change in state. Compatible problems can also be categorized by form of objective function, such as minimum or maximum of a sum or a product of the contributions from the individual stages, or by type of states. The state variables can be defined as discrete variables, continuous variables or vectors, and a problem can admit an infinite number of states for each stage.

### 2.5. An Issue in Populations Management

To illustrate the application of the previously described procedure, we consider at first an example already presented and analyzed, see [30]. It describes a predator–prey system modeling the interactions between the European red fox *Vulpes vulpes* and rodents as their prey. In addition, the parasitic tapeworm *Echinococcus multilocularis* may infect both populations. It is possible that by contact between rodents and domestic animals, also the latter may be infected by this parasite. As a consequence, it may ultimately be harmful to humans as well. Because the parasite is difficult to eradicate, a model to study its spread has been proposed. It concentrates only on the interactions in the wilderness, not explicitly modeling the parasites [30]. The system thus explicitly describes just the dynamics of the healthy foxes *F*, healthy rodents *S*, infected foxes *C* and infected rodents *I* populations.

Red foxes are the definitive hosts of the *Echinococcus multilocularis*. They become infected by preying on rodent intermediate hosts that ingested the parasite's eggs. The healthy rodents may get infected also by interactions with other infected rodents, or by direct contact with infected foxes, surviving their attacks, or indirectly, through fox-released environmental contamination.

From [30], the model is written using mass action disease incidence, meaning homogeneous populations mixing, which gives the following system

$$\frac{dF}{dt} = r(F + C) - mF + e(k_2 S + k_3 I)C + ek_1 FS - F(b_1 F + b_2 C) - \lambda FI - \alpha FC + \gamma_1 C, \quad (1)$$

$$\frac{dC}{dt} = \lambda FI + \alpha FC - (m + \mu)C - C(c_1 F + c_2 C) - \gamma_1 C,$$

$$\frac{dS}{dt} = s(S + I) - nS - S(g_1 S + g_2 I) - S(k_1 F + k_2 C) - \theta SI - \beta SC + \gamma_2 I,$$

$$\frac{dI}{dt} = \beta SC + \theta SI - I(n + \nu) - I[(g_3 S + g_4 I) + (\lambda F + k_3 C) + \gamma_2].$$

The model has been completely analyzed, determining its equilibria and their feasibility and stability conditions [30]. In particular, as a result of this investigation, a graph of the system stationary points, where the thriving populations are indicated without explicitly writing their equilibrium levels, linked by transcritical bifurcations, has been obtained. The parameters involved in the latter are the four populations' natural and disease-induced mortalities, $m$, $n$, $\mu$, $\nu$ for healthy foxes and rodents, and infected foxes and rodents, respectively. Figure 1 shows the graph of the equilibria and the transcritical bifurcations relating them.

In what follows, we assume that the current state of the system is the coexistence endemic equilibrium $(F, C, S, I)$, which is assumed to be stable. This corresponds to the actual conditions in the wild, where the two animal populations thrive, with the parasite being endemic among them. Our goal is to efficiently reach a pest-free equilibrium, either the parasite and rodent-free point $(F, 0, 0, 0)$, removing infected foxes and both healthy and infected rodents at the minimum cost. Alternatively, we will also seek the parasite-free equilibrium $(F, 0, S, 0)$, where the disease is eradicated, but healthy rodents thrive.

This problem can be rewritten as a minimum cost flow problem on the network built using the system equilibria as nodes: we consider the current state of the population system as the supply node and the desired state as the demand node. The arcs in the graph connect the equilibria that are linked by transcritical bifurcations. By acting on the bifurcation parameters, namely the mortalities, it would be possible to shift stability of one equilibrium and move to an adjacent node. Concretely, the operation of changing the parameter values requires some human operation. Therefore, we can associate a positive quantity to each arc, representing the cost of the corresponding activity. Our objective is to minimize the overall cost of the necessary operations needed to bring the system from its current status to the final desired configuration.

Since we set positive costs for each arc, we can assume for our model that there is just a unidirectional flow toward the demand node. This can be achieved by imposing that a backward step in a path before reaching the demand node would result in an increase in the total cost. We can therefore define a directed and connected network modeling a minimum cost flow problem, an instance of the shortest-path problem. The starting node and the demand node can be called origin and destination nodes. An image of the system graph is contained in Figure 1, where the arcs with label AM denote dependence on all four mortality parameters.

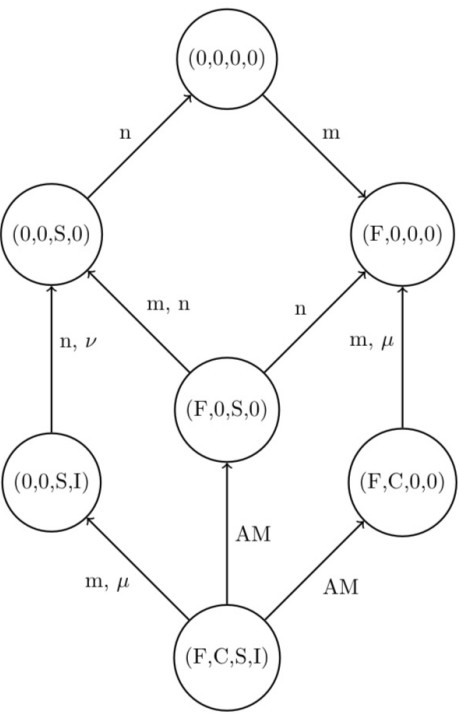

**Figure 1.** The graph of the equilibria for system (1) and the transcritical bifurcations relating them.

### 2.6. Algorithm and Implementation

The dynamic programming procedure is adapted to this shortest-path problem labeling the origin node 1 and the destination node $N$, given a connected graph containing $N$ nodes.

All the remaining nodes are arbitrarily labeled $2, \ldots, N - 1$. Each node corresponds to a state of the system. When implementing the algorithm, it is important to include a way to verify that this division in stages is preserved by the problem network. In this case, the principle of optimality necessary for a dynamic programming formulation can be applied. To this end, we employ a forward formulation of the Bellman–Ford algorithm [4,35–38].

### 2.7. Objective Function for the Bellman–Ford Algorithm

We use the following notation: in graph $G$, the indices $k$ and $j$ represent the nodes, $n$ denotes the current stage, $c_{ij}$ is the cost of the arc $ij$. The objective function $f_n^*(k)$ represents the length of a minimum path from the origin (node 1) to $k$ using at most $n$ arcs. It is defined through the recursive relationship:

$$f_n^*(k) = min_{j \in G}(f_{n-1}^*(j) + c_{kj}).$$

The initial condition is taken as $f_0^*(k) = 0$ if $k = 1$, and as $f_0^*(k) = \infty$ if $k \neq 1$.

This algorithm converges to the solution in $N - 1$ stages or less. An adapted backward formulation would be equivalent in terms of algorithm and results.

### 2.8. The Bellman–Ford Algorithm

To determine the optimal path for the problem illustrated in the previous section, we provide a MATLAB code. It represents an implementation of the Bellman–Ford algorithm, written in a form to be able to solve the general shortest-path problems.

**pmod** is the name of the main script. The number of nodes **n** and the costs matrix **C** are the only inputs required, where an edge is given infinite cost if not present on the graph.

The matrix below has been arbitrarily chosen to provide an example of code usage.

```
function [L,pmat] = pdmod()

n = 7;
C = inf(n,n);
C(2,1) = 4; C(1,3) = 2; C(1,4) = 2;
C(2,7) = 4; C(3,7) = 4; C(3,5) = 3;
C(4,5) = 3; C(5,6) = 5; C(6,7) = 4;
C(3,2) = 0;
```

We now describe the various segments of the driver code, sequentially implemented.

The vector **mincost** represents the objective function, where at stage $k$, the element $i$ stands for the optimal value of a path from the origin to the node labeled $i$, composed of at most $k$ arcs.

```
mincost = inf(1,n);
mincost(1) = 0;
neg = inf(1,n);
```

**pmat** is the matrix of the optimal paths. Taking any one of its rows, the $i$th element is the label of the node that immediately precedes node $i$ in the optimal node sequence.

```
q = zeros(1,n);

p = inf(1,n);
p(1) = 1;
pmat = [];
```

**neg** will be used to check for negative cycles once $n$ stages have been completed. The script **ofu**, shown in the next section of code, is then called $n$ times in order to update the vector **mincost** and **pmat**. The optimal value is saved as **L**.

```
for stages = 1:n
if stages == n
neg = mincost;
end % if stages
[pmat,p,mincost] = ofu(n,C,mincost,stages,p,pmat);
end % for stages
if neg ~= mincost
sprintf('negative cycle(s)')
return
end % if neg

L = mincost(n);
```

The objective function is updated by scanning the network arcs one by one. Once the last necessary stage, $n - 1$, is reached, the code constructs the matrix **pmat** containing the optimal paths by scanning the function values. The matrix is filled one row at a time, filling the vector $p$ containing the positions of the nodes in the optimal sequence by comparing the values of **mincost**, and saving it as a row vector upon completion of the procedure.

```
function [pmat,p,mincost] = ofu(n,C,mincost,stages,p,pmat)
for i = 1:n
for j = 1:n
if C(i,j) < inf
if mincost(j) >= mincost(i) + C(i,j)
mincost(j) = mincost(i) + C(i,j);
if stages == n
p(j) = i;
```

```
g = n;
while g > 1
if p(g) == 1
pmat = [pmat;p];
end % if p(g)
if p(g) < inf
g = p(g);
else g = 1;
end % if p(g)
end % while g
end % if stages
end % if mincost(j)
end % if C(i,j)
end % for j
end % for i
```

The following section of the main code builds the adjacency matrix of the system graph and scans the matrix of the optimal paths **pmat** from the origin to the destination, producing visual outputs and deleting redundant rows leading to already recorded paths. Here, **jq** and **iq** are the vectors containing the start and end nodes of the edges to mark as part of an optimal path. **q** is a new path from the origin to the destination.

```
A = false(n,n);
for i = 1:n
for j = 1:n
if C(i,j) < inf
A(i,j) = 1;
end
end
end
G = digraph(A);

k = 1;
while k <= size(pmat,1)
jq = [];
iq = [];
j = n;
while j ~= 1
jq = [jq,j];
q(j) = pmat(k,j);
j = q(j);
iq = [iq,j];
end
```

The section of above code highlights in red an optimal path on the system graph, and in yellow the origin and the destination. The result is in the following figure.

```
figure()
F = plot(G,'lineWidth',6.7,'Markersize',6.7);
highlight(F,iq,'Nodecolor','r')
highlight(F,iq,jq,'LineStyle',':','Edgecolor','r')
highlight(F,[1,n],'Nodecolor','y')
```

Any rows representing duplicate paths are deleted from **pmat**, and counters increase according to size corrections.

```
c = 1;
while c <= size(pmat,1)
qpar = true;
for g = 1:length(q)
if q(g) ~= 0 && q(g) ~= pmat(c,g)
qpar = false;
break
end
end
if c ~= k && qpar
pmat(c,:)  = [];
k = k+1;
end

c = c+1;
end
k = k+1;
end
```

## 3. Results

### 3.1. Various Examples of Model (1)

We now present some examples of the implementation of the code. At first, we consider different situations for the model (1), varying the weights of the various arcs to show the versatility of the code. We then implement it on other models of similar nature that are not described in detail, but that have already been analyzed and published.

Figure 1 comes from the paper [30], as do Figures 2–7. Figures 8, 9 and 10 instead are, respectively, from the other references [31], [32] and [33]. As such, they are starting points for our implementation here, and need no further justifications. To deepen the information about the models from which the graphs are taken, it is enough to consult the appropriate cited references. Note that in the ecosystem related to Equation (1), predators survive in the absence of the rodents because it is assumed that foxes survive on many preys other than the rodent population modeled in the system (1).

#### 3.1.1. Attaining the Parasite and Rodent-Free Equilibrium

Consider model (1), where the equilibria are given in Figure 1. It has $n = 7$ equilibrium states. It is depicted in Figure 2, where the states are indicated by numbers and correspond to the equilibria of Figure 1, with the convention that the origin of the graph is always labeled node 1 and the destination is always labeled as node 7. Hence, the coexistence endemic equilibrium $(F, C, S, I)$ corresponds to the origin at the bottom of the graph, and the destination for this example is the parasite and rodent-free point $(F, 0, 0, 0)$. Note that in Figure 2, these points occupy the same positions as in Figure 1. For the remaining nodes, we have the following matches reported in Table 1.

**Table 1.** Pairings of the states in the network with the ecosystem equilibria.

| Node | Equilibrium |
| --- | --- |
| 2 | $(F, C, 0, 0)$ |
| 3 | $(F, 0, S, 0)$ |
| 4 | $(0, 0, S, I)$ |
| 5 | $(0, 0, S, 0)$ |
| 6 | $(0, 0, 0, 0)$ |

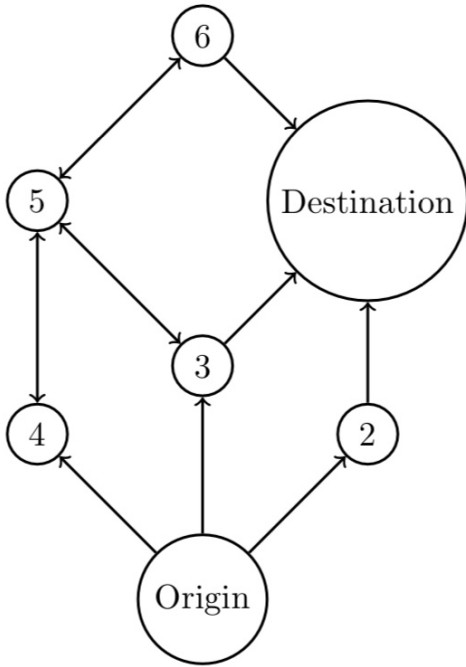

**Figure 2.** The graph with labeled nodes corresponding to the one in Figure 1.

We give two different examples, using different costs (weights) for the various arcs, to show the performance of the algorithm. After providing the costs matrix, we give the optimal value and the optimal path as a sequence of the nodes to be attained at each step. The minimum cost path is also shown in the figures in red color.

**Case 1**

Costs:

```
C(1,2) = 28; C(1,3) = 51; C(1,4) = 16;
C(2,7) = 42; C(3,7) = 12; C(3,5) = 46;
C(4,5) = 14; C(5,6) = 42; C(6,7) = 29;
C(5,3) = 20; C(5,4) = 37; C(6,5) = 12;

Optimal value:  62
Optimal path:   (1,4,5,3,7), see Figure 3.
```

**Case 2**

Costs:

```
C(1,2) = 8; C(1,3) = 51; C(1,4) = 16;
C(2,7) = 42; C(3,7) = 12; C(3,5) = 46;
C(4,5) = 14; C(5,6) = 11; C(6,7) = 9;
C(5,3) = 20; C(5,4) = 37; C(6,5) = 12;

Optimal value:  50
Optimal paths:  (1,2,7), (1,4,5,6,7), see Figure 4. Here the path is not unique.
```

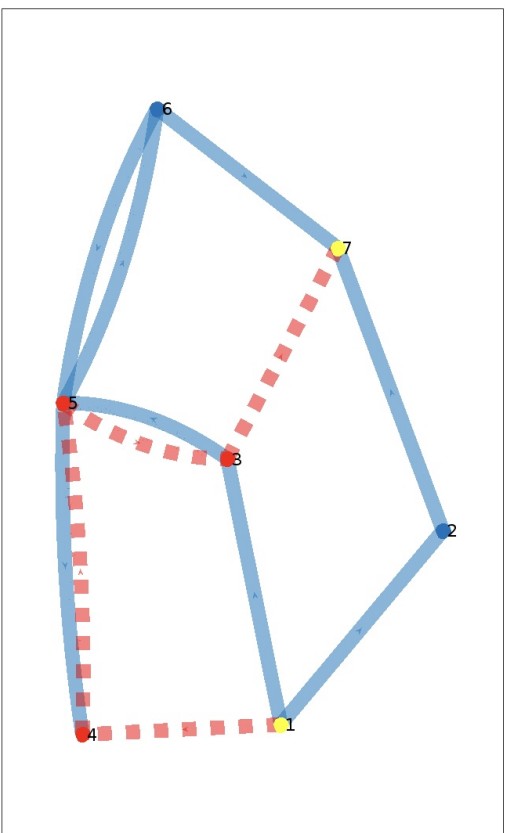

**Figure 3.** The optimal path (in red) corresponding to Case 1.

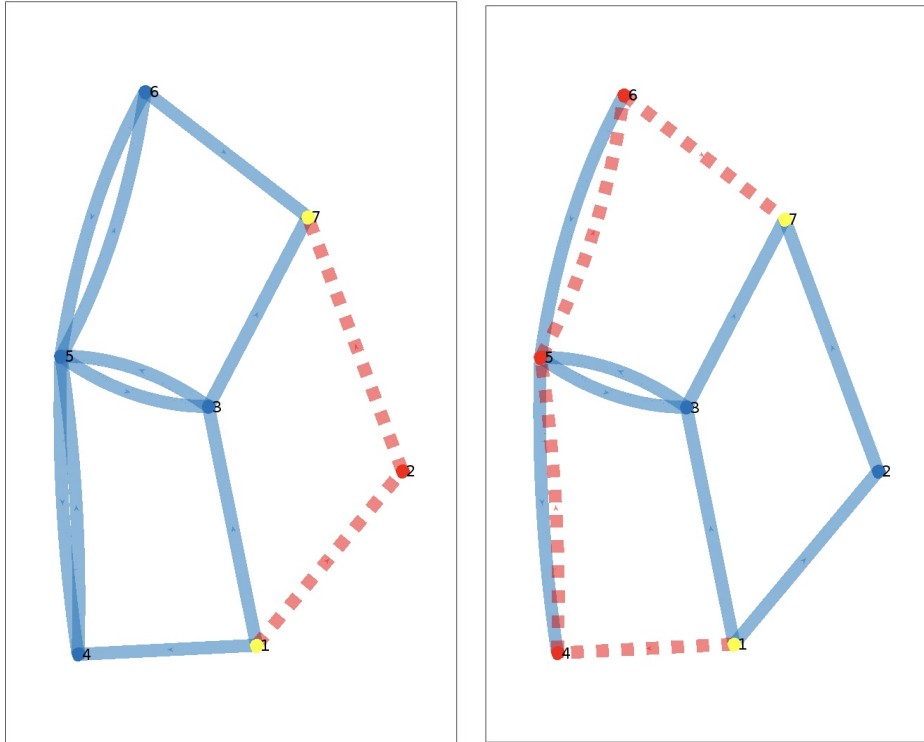

**Figure 4.** (**Left**): the optimal path (in red) corresponding to Case 2. (**Right**): the alternative optimal path through the sequence of nodes (1,4,5,6,7).

### 3.1.2. Attaining the Parasite-Free Equilibrium

In this case, we keep the same origin, the coexistence state, but we seek to optimally attain the state $(F, 0, S, 0)$ corresponding to node $n = 3$, Figure 5.

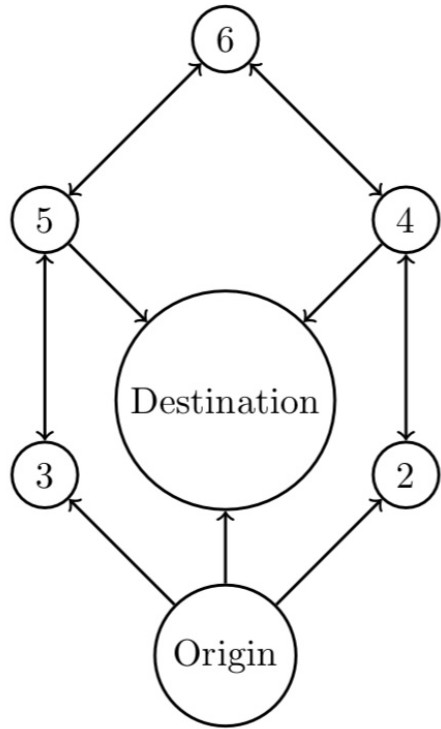

**Figure 5.** The graph corresponding to the problem of attaining the parasite-free equilibrium.

**Case 3**

Costs:

```
C(1,2) = 4; C(1,3) = 13; C(1,7) = 24;
C(2,4) = 2; C(3,5) = 4; C(4,6) = 6;
C(5,7) = 2; C(5,6) = 10; C(4,7) = 18;
C(6,4) = 9; C(4,2) = 12; C(6,5) = 4;
C(5,3) = 20;
```

```
Optimal value:  18
Optimal path:   (1,2,4,6,5,7), see Figure 6.
```

Costs:

```
C(1,2) = 4; C(1,3) = 10; C(1,7) = 24;
C(2,4) = 2; C(3,5) = 4; C(4,6) = 16;
C(5,7) = 3; C(5,6) = 10; C(4,7) = 11;
C(6,4) = 9; C(4,2) = 12; C(6,5) = 14;
C(5,3) = 20;
```

```
Optimal value:  17
Optimal paths:  (1,3,5,7), (1,2,4,7), see Figure 7.
```

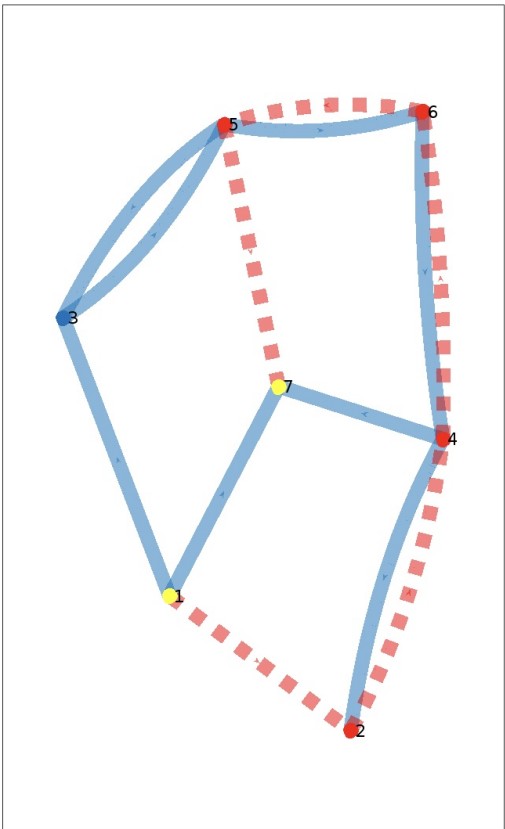

**Figure 6.** The optimal path (in red) corresponding to Case 3.

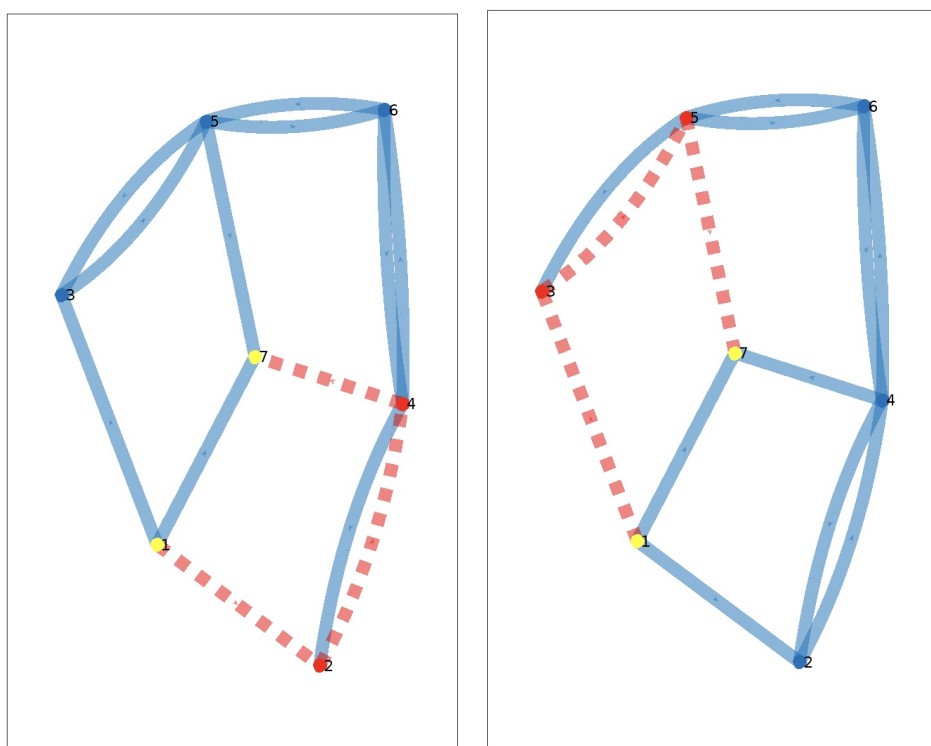

**Figure 7.** (**Left**): the optimal path (1,3,5,7) (in red) corresponding to Case 4. (**Right**): the alternative optimal path through the sequence of nodes (1,2,4,7).

### 3.2. Examples of Applications to Other Ecological Models

We start by considering the model investigated in [31], with the ecosystem's equilibria visualized in the left frame of Figure 8. Here we have

Costs:

```
C(1,2) = 50; C(1,3) = 44; C(1,8) = 167; C(2,8) = 110;
C(6,8) = 89; C(4,8) = 69; C(2,4) = 31; C(4,2) = 77;
C(2,5) = 33; C(5,2) = 37; C(2,5) = 51; C(5,3) = 86;
C(3,6) = 98; C(6,3) = 87; C(4,7) = 24; C(7,4) = 15;
C(5,7) = 13; C(7,5) = 22; C(7,6) = 123; C(6,7) = 132;

Optimal value:  150
Optimal path:   (1,2,4,8), see the right frame of Figure 8.
```

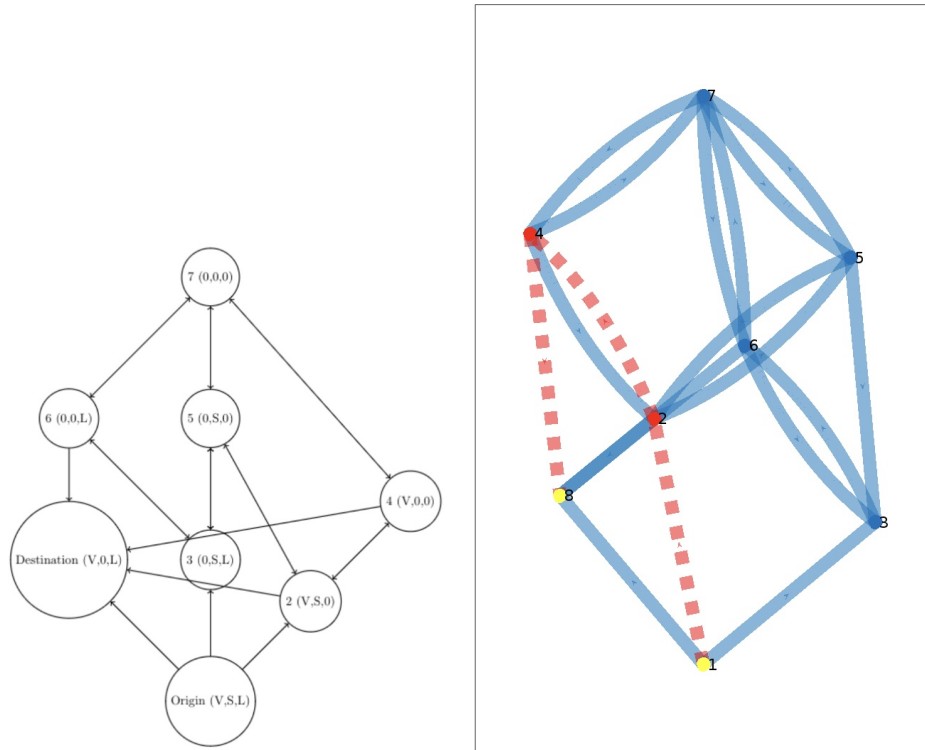

**Figure 8.** (**Left**): the graph of the ecosystem's equilibria of [31]. (**Right**): the optimal path (1,2,4,8).

Next, we consider the model introduced in [32]. The ecosystem's equilibria are shown in the left frame of Figure 9, and the optimal solution is in the right frame. Here we have

Costs:

```
C(1,2) = 23; C(1,3) = -2; C(1,10) = 46; C(1,5) = 4;
C(3,7) = 18; C(7,3) = 18; C(3,4) = 11; C(4,3) = 10;
C(3,10) = 31; C(2,4) = 5; C(4,2) = 12; C(2,8) = 20;
C(8,2) = 24; C(2,6) = 31; C(6,2) = 11; C(4,8) = 21;
C(8,4) = 3; C(4,9) = 16; C(9,4) = 4; C(5,7) = 10;
C(7,5) = 11; C(5,6) = 12; C(6,5) = 8; C(9,8) = 9;
C(8,9) = 10; C(9,7) = 4; C(7,9) = 14; C(9,6) = 21;
C(6,9) = 45; C(7,10) = 2; C(8,10) = 6;

Optimal value:  16
Optimal path:   (1,5,7,10).
```

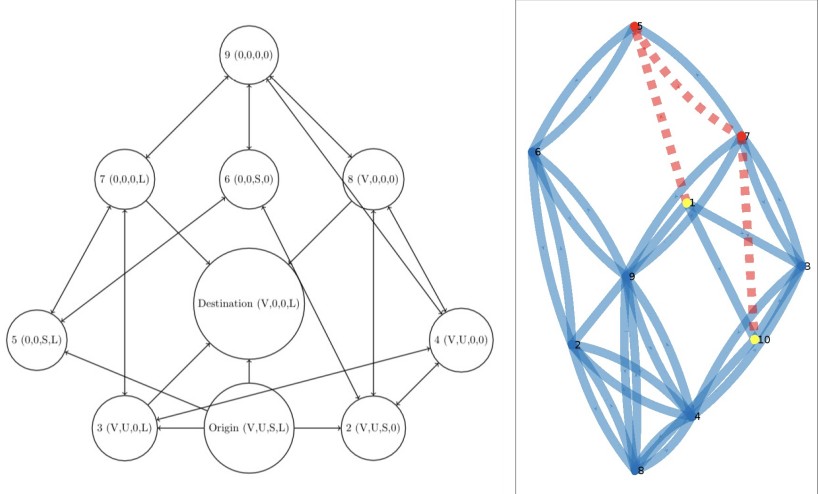

**Figure 9.** (**Left**): the graph of the ecosystem's equilibria of [32]. (**Right**): the optimal path (1,5,7,10).

We finally consider the model of [33], shown in the left frame of Figure 10.
Costs:

```
C(1,2) = 16; C(1,3) = 11; C(1,4) = 4; C(2,12) = 5;
C(8,12) = 3; C(9,12) = 2; C(5,2) = 3; C(5,8) = 7;
C(3,5) = 15; C(4,5) = 4; C(2,6) = 4; C(6,2) = 16;
C(6,4) = 5; C(4,7) = 12; C(3,7) = 8; C(7,10) = 6;
C(10,6) = 2; C(5,10) = 4; C(10,11) = 2; C(11,10) = 5;
C(11,9) = 6; C(6,9) = 7; C(8,11) = 3;

Optimal value:  16
Optimal path:  (1,4,5,2,12).
```

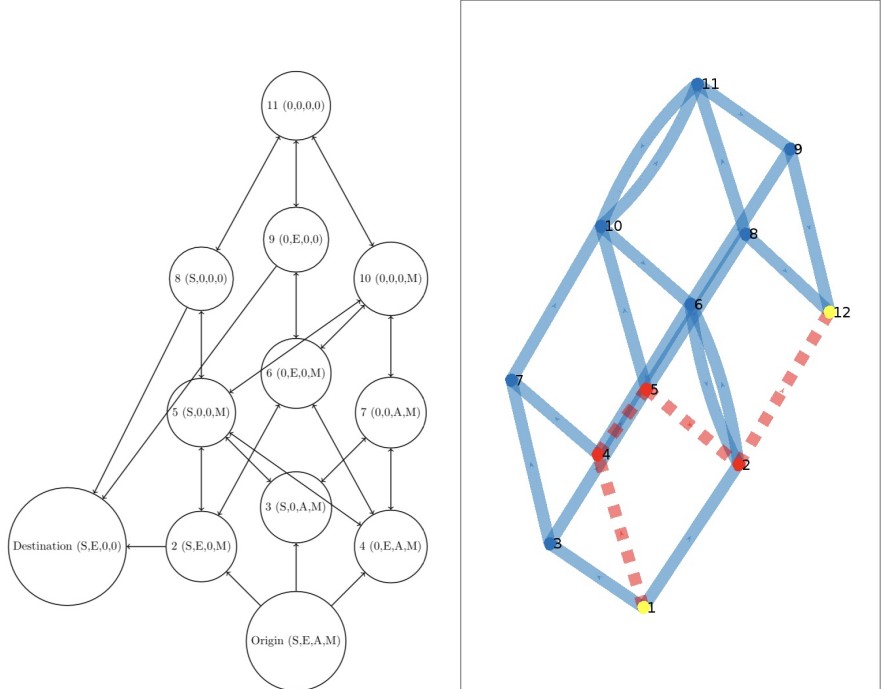

**Figure 10.** (**Left**): the graph of the ecosystem's equilibria of [33]. (**Right**): the optimal path (1,4,5,2,12).

## 4. Discussion

We presented an algorithm to automatically detect the optimal way of attaining a certain desired equilibrium in an ecological setting where pests or pathogenic agents are present and should be controlled or eradicated. The algorithm is sufficiently general to be adapted to other cases. In the previous section, we provided some other examples where the models come from the recent literature.

The algorithm is versatile because it requires only the knowledge of the graph relating to the various system's equilibria, which is the result of a previous necessary analytical study. The ecological part concerns the weights to be given to the various arcs, and this comes only from information that is specific to each particular issue at hand.

A number of different applications have been shown, using four models in the current literature. It is not possible to compare their results because, as mentioned above, the cost of each arc in the various models is completely hypothetical, and the output information provided by the code depends heavily on this choice. Therefore, general conclusions cannot be drawn. The examples shown in Section 3 were provided mainly to show the versatility of this approach. However, in specific instances, the algorithm provides useful information for the ecological manager because, in these instances, he most likely will be able to provide meaningful values for the operating costs that represent essential parameters in the scheme. The fact that the result depends on the employed weight values is shown in Figure 7, where two different solutions to the same problem appear for different costs associated with the same graph.

## 5. Conclusions

To sum up, we presented an algorithm to determine the way of altering the outcome of an ecosystem at a minimal cost.

This, to our knowledge, is the first application of dynamic programming techniques to ecological settings.

The solution relies on two major points that also represent the limitations of this approach: the whole set of the system's equilibria must be known, together with the transcritical bifurcations that link them. As mentioned in the Introduction, at the moment, this approach cannot be applied in particular if the dynamical system exhibits persistent oscillations or chaotic attractors. To deal with these problems, further work is necessary to be performed in the future. In addition, reliable estimates of the costs of the actions for changing the system parameters to push them over the critical thresholds, so that the ecosystem moves to an adjacent equilibrium, must be known. In case they are not exactly known, a repeated application of this algorithm perturbing these values could be used to assess an optimal (or suboptimal) solution anyway. Overall, to our knowledge, this represents a novel and useful application for the Bellman–Ford algorithm.

It provides the manager of the ecosystem useful information on how to act on the current situation, e.g., a pest-infected crop, and change it to the desired parasite-free equilibrium outcome. Above all, since there are explicit and hidden costs associated with all the management options, the algorithm indicates a minimum cost solution, within the natural constraints of the ecosystem.

**Author Contributions:** Conceptualization, E.V.; Methodology, E.V.; Software, A.R.; Validation, A.R.; Investigation, A.R.; Writing—original draft, A.R.; Writing—review & editing, E.V.; Supervision, E.V. All authors have read and agreed to the published version of the manuscript.

**Funding:** This work was partially supported by the local research project "Metodi numerici per l'approssimazione e le scienze della vita" of the Dipartimento di Matematica "Giuseppe Peano", Università di Torino.

**Data Availability Statement:** Not applicable.

**Acknowledgments:** The authors thank the referees for their constructive comments.

**Conflicts of Interest:** The authors declare no conflict of interest.

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
