# Peer review of "A Dynamic Programming Approach to Ecosystem Management"

_algorithms, doi:10.3390/a16030139_

Round 1

Reviewer 1 Report

In this work, the authors propose a solution to the problem of modeling ecosystems when they have multiple steady states and in which transcritical bifurcations appear. Using dynamic programming tools, the authors claim that they can map the equilibria of the system and detect the cheapest path to reach the desired objective. In this manuscript, an algorithm for the solution of this problem is presented. This work needs to be improved for publication, especially in its introduction.

I recommend the following points to be taken into account.

1.- The introduction should be improved, citing previous works by different authors that are relevant to the investigation. The novelty or contribution of the work should also be highlighted. In addition, the first references in the text appear with the numbers 3 and 5, in any case they should be 1 and 2. 

2.- I recommend separating the Discussion of the results from the conclusions. The authors should add an extra point with the conclusions indicating or affirming the contributions of the work.

3.- It would be interesting (if possible) to compare the results of this algorithm with other types of solutions for the same problem.

4.- The references should be expanded and updated.

Author Response

1.- The introduction should be improved, citing previous works by different authors that are relevant to the investigation. The novelty or contribution of the work should also be highlighted. In addition, the first references in the text appear with the numbers 3 and 5, in any case they should be 1 and 2. 

ANSWER 
The introduction has been expanded and many more citations are provided.

2.- I recommend separating the Discussion of the results from the conclusions. The authors should add an extra point with the conclusions indicating or affirming the contributions of the work.

ANSWER 
Written two separate brief sections

3.- It would be interesting (if possible) to compare the results of this algorithm with other types of solutions for the same problem.

ANSWER 
This is not currently possible, as this problem, of stepping through the equilibrium points of a dynamical system in the context
of ecology, to our knowledge has not been addressed an no other such papers exist in the current literature.
There are some that address continuous control problems of dynamical systems, but they do not address the same issue.
The reference by Lenhart is cited anyway.

4.- The references should be expanded and updated.

ANSWER 
Done, many more papers and books on the topic are now cited.

Reviewer 2 Report

The article presented deals with an interesting subject but has a number of important shortcomings that should have been corrected before submission to the journal. From my point of view they are of such importance that they prevent publication. The following is a list of some of the defects found:

1. Most equations are not numbered.

2. The references appear disordered throughout the text, and some of them are not found.

3. References are scarce and insufficient. To make matters worse, most of them refer to the work of one of the authors. This could be justified if it were an absolutely leading and original topic, but as it appears in the article, it seems rather a lack of rigor in the search for previous research.

4. The introduction is extremely brief and lacks serious and in-depth discussion. In general, the authors' proposal should be justified and based on rigorous studies. In general, ecological systems and the associated equations present nonlinearities and often chaotic character, so that the possibility of management by modifying parameters would only make sense for oversimplified models of reality (and therefore without utility) or systems with adequate characteristics (such as a set of attractors that allow such interaction).

5. The results in Figure 1 should be justified. How can a model in which predators feed on prey have equilibrium points in which only predators exist? Either the analysis of the results or the equations are inconsistent.

6. Sections 4.1 to 4.3: The contents of these sections should be in the methodology.

7. The complete and detailed code is unnecessary. The pseudocode that allows understanding the algorithm would be sufficient.

8. Section 5.1 should be in results.

9. The results are presented in a simplified and unclear manner. Discussions regarding the nature of the different costs and the environmental feasibility and consequences should be included.

Author Response

1. Most equations are not numbered.

ANSWER 
We numbered only the relevant ones, that are possibly cited in the body of the paper. For the other ones this is not necessary,
nor the usual practice in a mathematical context.

2. The references appear disordered throughout the text, and some of them are not found.

ANSWER 
Done, many more papers and books on the topic are now cited and reordered.

3. References are scarce and insufficient. To make matters worse, most of them refer to the work of one of the authors. This could be justified if it were an absolutely leading and original topic, but as it appears in the article, it seems rather a lack of rigor in the 
search for previous research.

ANSWER 
Done, many more papers and books on the topic are now cited and reordered.
Our own references are necessary as the examples discussed in the paper rely on them.

4. The introduction is extremely brief and lacks serious and in-depth discussion. In general, the authors' proposal should be justified and based on rigorous studies. In general, ecological systems and the associated equations present nonlinearities and often chaotic character, so that the possibility of management by modifying parameters would only make sense for oversimplified models of reality (and therefore without utility) or systems with adequate characteristics (such as a set of attractors that allow such interaction).

ANSWER 
The introduction has been expanded and many more citations are provided.
As far as ecological systems are concerned, chaos is not the only behavior of such systems,
and is excluded from these considerations. This is stated in the second paragraph of the introduction.

5. The results in Figure 1 should be justified. How can a model in which predators feed on prey have equilibrium points in which only predators exist? Either the analysis of the results or the equations are inconsistent.

ANSWER 
Figure 1 comes from a paper cited in the references, as do Figures 2, 5, 7, 8 and 9.
As such they are starting points for our implementation here, and need no further justifications.
If the reader wants to deepen his/her knowledge, s/he can consult the appropriate cited references.
And the remark on how predators survive would simply been answered by the referee himself, if he
consulted the cited paper. In it, it is assumed, as it is the case, that foxes survive on many prey other
than the rodents population modeled in the system (1).

6. Sections 4.1 to 4.3: The contents of these sections should be in the methodology.

ANSWER 
Done sections moved

7. The complete and detailed code is unnecessary. The pseudocode that allows understanding the algorithm would be sufficient.

ANSWER 
The journal directives seem to emphasize that the algorithmic details are put in the paper. Hence we leave it.

8. Section 5.1 should be in results.

ANSWER 
Done section moved

9. The results are presented in a simplified and unclear manner. Discussions regarding the nature of the different costs and the environmental feasibility and consequences should be included.

ANSWER 
The cost issue is addressed in the discussion section.

Reviewer 3 Report

The authors propose a model for ecosystem using dynamic programming. The problem is divided to sub-problems and the results of every stage is used in the next one.The proposed model and algorithms are tested on examples from other authors, related with ecosystems. The paper is well written and the proposed model and algorithm are described in details. The implementation is illustrated by small examples.

The weakness of the paper is reference list. It is too short.

Author Response

The weakness of the paper is reference list. It is too short.

ANSWER 
Done, many more papers and books on the topic are now cited.

Reviewer 4 Report

The first two sections have very high similarity to previously published work. Needs to rewrite and the can be reconsidered for publication. Please find the similarity report. 

Author Response

The first two sections have very high similarity to previously published work. Needs to rewrite and the can be reconsidered for publication. Please find the similarity report. 

ANSWER 
These sections are meant only for the readers that are unfamiliar with the topic. They are based on definitions that are
standard in the literature, and one does not want to mess with standard definitions even though they may imply similarities
with other works (I would say textbooks). We stated that clearly at the start of the section, and for this reason the rest
remains unchanged.

Round 2

Reviewer 2 Report

The content of the article has improved significantly. However, some remarks should be considered again:

+ The result in Figure 1 can be explained briefly by the authors as they have done in the responses to the review, thus avoiding that any potential reader of their article who is curious should go to the previous work to clarify such a minor point.

+ The conclusions section should include a reflection on the limitations of the proposed methodology and the future lines that can be followed from this  work.

+ With respect to the inclusion of code, the emphasis referred to by the authors is usually sufficiently resolved by including adequate pseudocode. It is always more elegant and useful to readers than the details of the concrete implementation in the environment and language used by the authors. However, there is no point in having a discussion about this, as it is something that should be decided by the editor.

Author Response

+ The result in Figure 1 can be explained briefly by the authors as they have done in the responses to the review, thus avoiding that any potential reader of their article who is curious should go to the previous work to clarify such a minor point.

ANSWER Done see page 10

+ The conclusions section should include a reflection on the limitations of the proposed methodology and the future lines that can be followed from this  work.

ANSWER Done see page 17

+ With respect to the inclusion of code, the emphasis referred to by the authors is usually sufficiently resolved by including adequate pseudocode. It is always more elegant and useful to readers than the details of the concrete implementation in the environment and language used by the authors. However, there is no point in having a discussion about this, as it is something that should be decided by the editor. 

ANSWER We now leave the code, in case the Editor asks us to write instead the pseudocode we will insert it.

Reviewer 4 Report

Showed great improvement and can be considered for publication. Thank you for all the efforts

Author Response

Thanks for your report.

The authors

Round 3

Reviewer 2 Report

Since the suggestions have been mostly covered by the authors, we leave the difference of opinion in the hands of the editor and since this is a matter that concerns a non-essential aspect of the article, I modify my advice regarding the article accordingly.

Author Response

Thanks, as I understand it, we should receive a communication by the Editor in case he wants only the pseudocode instead of the whole program.